# Clinical and laboratory comparison of severe (Group B and C) Dengue cases with molecular characterization from 2019 epidemics in Dhaka, Bangladesh

**Fazle Rabbi Chowdhury**[1,2]*, **Zazeba Hossain**[1], **Nahid Parvez**[3], **Forhad Uddin Hasan Chowdhury**[4], **Mohammad Anwarul Bari**[5], **Sudip Ranjan Deb**[6], **Mustak Ibn Ayub**[3], **Md Uzzwal Mallik**[4], **Sakib Aman**[4], **Mohammad Ahsanul Haque**[5], **Md Arman Hossain**[4], **Murada Alam**[5], **Muhammad Kamrul Islam**[6], **Md Mujibur Rahman**[4]

1 Department of Internal Medicine, Bangabandhu Sheikh Mujib Medical University, Dhaka, Bangladesh,
2 Department of Tropical Medicine, Mahidol-Oxford Tropical Medicine Research Unit (MORU), Bangkok, Thailand, 3 Department of Genetic Engineering and Biotechnology, Dhaka University, Dhaka, Bangladesh, 4 Department of Medicine, Dhaka Medical College and Hospital, Dhaka, Bangladesh, 5 Department of Medicine, Sir Salimullah Medical College and Mitford Hospital, Dhaka, Bangladesh, 6 Department of Medicine, Mugda Medical College and Hospital, Dhaka, Bangladesh

* masterfazlerabbi@gmail.com

**Data Availability Statement:** The authors confirm that all data underlying the findings are fully

## Abstract

Acute arboviral infections like dengue have a significant negative socioeconomic and health impact on many tropical and subtropical areas of the world. About 3.9 billion Individuals are at risk of contracting the dengue virus and Asia bears the brunt of that load. Bangladesh, like other south-east Asian countries faced a massive outbreak of dengue in 2019. This cross sectional study was done in three tertiary care centers in Dhaka, Bangladesh during this worst outbreak of dengue. The study was conducted from 1st July to 31st December, 2019 with an aim to describe the clinical and laboratory variations among severe dengue cases and to conduct a serotype survey. This might help to understand the future changes in the clinical or serological profile of the circulating dengue virus. The study enrolled 1978 participants who were grouped into group B (Patients with warning signs or risk factors who should be admitted for close observation as they approach critical phase) and C (Patients having severe plasma leakage leading to dengue shock and/or fluid accumulation with respiratory distress, severe organ impairment and severe metabolic abnormalities) according to national guidelines. Furthermore, 81 samples were serotyped using Qiagen One step RT-PCR kit (Cat. No: 210212). In addition sequencing (ABI sequencing platform) of partial C-prM gene of five DENV-3 isolates were done and analyzed (BLAST tool of NCBI) for phylogenetics (MEGA6 software package). Among the 1978 enrolled participants group B and C patients were 1580 (80%) and 398 (20%) respectively. The median (IQR) age of the patients were 26(11 to 41). Maximum proportion of the enrolled were male (72.3%) (p = 0.0002). Most common co- morbidities were hypertension (90; 4.5%) and DM (70; 3.5%). Group C patients more commonly presented with vomiting (p 0.133), diarrhea (p<0.0001) and abdominal pain (p 0.0203). The common mode of bleeding was melaena (12%). Thirteen (0.7%) patients succumbed to death, 12 of them belonged to group B who mostly presented

available without restriction at figshare, the link is given below- https://figshare.com/articles/dataset/Dengue_Serotype_and_Genotype_2019_Dhaka_outbreak/24686820 DOI: https://doi.org/10.6084/m9.figshare.24686820.v1.

**Funding:** We acknowledge financial support received by MMR from Bangladesh Medical Research Council, Mohakhali, Dhaka-1212, Bangladesh (2019-2020/605). The funders had no role in study design, data collection and analysis, decision to publish, or preparation of the manuscript.

**Competing interests:** The authors have declared that no competing interests exist.

with GI manifestations (99.9%) and melaena (12%). Only 5.6% of the cases were secondarily infected. In group C cases low hemoglobin and hematocrit was observed with high AST (p 0.004, 0.006 and 0.0016 respectively). Fluid requirement was also more in the same group (p<0.0001). Group B patients had a higher platelet requirement (p = 0.0070). Twelve patients (0.7%) required ICU. The management profile of these cases are showcased here which highlights minimal use of antibiotics and no application of steroids, which abides by the current national protocol. Furthermore, 81 samples from enrolled participants were serotyped and majority (79%) yielded DENV-3, followed by DENV 2 & 3 co-infection (13.6%) and DENV-2 alone (7.4%). Following phylogenetic analysis DENV-3 and DENV2 were deemed to be of genotype I and cosmopolitan variety respectively. This study presents the first instance of heterogeneous co-infection with several serotypes since 2000 in Bangladesh. It also gives an overview of serotype prevalence, management evaluation and clinical results that promises to navigate future control planning.

## Author summary

Bangladesh experienced an unprecedented dengue surge in 2019. This cross sectional study was done in 3 hospitals in Dhaka, Bangladesh spanning from 1st July to 31st December, 2019 enrolling a total of 1978 participants. We observed a shift of clinical presentations and varied laboratory alterations in the severe groups which was portrayed according to national guideline case definitions. Most common co- morbidities were hypertension (4.5%). The major mode of bleeding was melaena (12%). Total 13 (0.7%) patients succumbed to death. The management received by these patients were also elaborated. Although Bangladesh is a hyper endemic region, the serotype survey is rarely done because of logistic and financial constraints. However it has become a burning need to study the infecting serotype of the severe cases to help navigate the control measures and predict the atypical manifestations and also for future vaccine deployment. Here, 81 samples were serotyped and most were serotyped as DENV-3 (79%), followed by DENV 2 (7.4%). Interestingly we also found cases co infected with these two serotypes (13.6%). These strains were sequenced and further molecular analysis were performed. This study presents the first instance of heterogeneous co-infection with several serotypes since 2000 in Bangladesh.

## Introduction

Dengue is the most rapidly spreading mosquito-borne viral illness transmitted to humans by infected *Aedes* mosquitoes. Principally by *Aedes aegypti* and less often by *Aedes albopictus* [1]. It belongs to Flaviviridae family containing closely related four serotypes (DENV 1–4) which are further classified into different genotypes. These serotypes can co-circulate and many countries are hyper endemic for all four serotypes. Recovery from one serotype provides life-long immunity against that serotype only. Although there is transitory and partial cross-immunity due to antibody-dependent enhancement (ADE), secondary infection with an additional serotype raises the risk of developing severe dengue [2]. Before 1970 only nine (9) countries had reported severe dengue epidemics, now the disease has spread its endemicity in more than 100 countries worldwide. The number of dengue cases reported to WHO increased over

8 fold over the last two decades [2]. It is predicted that by 2080, 60% of the world population (over 6.1 billion) will be at risk of dengue infection [3].

Bangladesh encountered first official outbreak in 2000 reporting 5551 cases with 1.6% case fatality. Since then there has been cyclical outbreaks every few years [4–7]. Largest dengue upsurge till date was recorded in 2019 globally. That year the Americas reported 3.1 million cases. In Asia, Malaysia followed by Philippines and Vietnam reported 1,31,000, 4,20,000 and 3,20,000 cases respectively [2]. Similar phenomenon reported in Nepal [8]. Bangladesh also faced its highest dengue cases throughout the country that year. About 1,01,354 (one hundred one thousand, three hundred fifty-four) confirmed cases required admission and 164 deaths reported [9]. The blow of dengue in 2019 was not just restricted to endemic cities but was spread nationwide [10].

The multi-annual fluctuations in DF (Dengue Fever) or DHF (Dengue Hemorrhagic Fever) incidence as well as cyclical oscillations of the individual serotypes, where the dominant type is gradually replaced over time, are indicative of the temporal epidemiological pattern of dengue [11]. These shift in serotypes and genotypes may lead to severe dengue outbreaks due to ADE [12].

Despite being frequently afflicted with dengue outbreaks, Bangladesh does not have adequate empirical study on dengue virus serotype detection, severity correlation and molecular evolution. This study was done in 2019 during the first large outbreak in Dhaka, which is known as the epicenter of dengue outbreak [13], with an aim to explore the clinical presentations and outcome of the severe cases along with evaluating the management they received. This study also stepped up to detect the prevalent serotype with its phylogenetic analysis.

## Materials and methods

### Ethical approval and study enrollment profile

Written informed consent was taken from every patient or lawful guardian before enrollment. Counselling was done about the nature of the study and about collection of serum sample. Ethical clearance was obtained from Ethical review committee of Bangladesh Medical Research Council, Mohakhali, Dhaka-1212, Bangladesh (memo no-RP/2019/365).

This cross-sectional study done from 1st July to 31st December, 2019 at three tertiary care government hospitals in Dhaka, Bangladesh namely, Dhaka Medical College and Hospital, Mugda Medical College Hospital and Sir Salimullah Medical College Hospital. The whole study was coordinated by Bangabandhu Sheikh Mujib Medical University (BSMMU). S1 Fig shows the map and enrollment profile of the study sites. Suspected dengue cases (Fever with any two of nausea/vomiting, rash, aches/pain, leucopenia, any warning sign or positive tourniquet test) [14] were screened. Those who had fever for 2–7 days, aged more than 16 years and showed either NS1 antigen positivity or Immunoglobulin M positivity were included in this study. Those who did not wish to participate and were co-infected with any bacterial or parasitic infection were excluded from this study. Patients who were complicated into "Expanded dengue syndrome" [14] were also excluded from this study. After enrollment they were broadly divided into Group B (Patients with warning signs [14] or risk factors who should be admitted for close observation as they approach critical phase) and Group C (Patients having severe plasma leakage leading to dengue shock and/or fluid accumulation with respiratory distress, severe organ impairment and severe metabolic abnormalities) according to national dengue case management guidelines [14]. Primary infection were reported by laboratory evidence of NS1 or Anti dengue IgM RDT (rapid diagnostic test) and secondary infections were declared when patients had Anti dengue IgG RDT positivity. Probable and confirmed dengue cases were reported by antidengue IgM positivity and NS1 antigen detection respectively. Data

input, sample collection and tests were done following a predefined protocol detailed in S1 text (S1.1 and 1.2).

## Dengue virus serotyping

Serotyping and genotyping was done in the laboratory of the Department of Genetic Engineering and Biotechnology, University of Dhaka, Bangladesh. Reverse transcriptase based nested PCR was used for serotyping following the manufacturers protocol (Qiagen One step RT-PCR kit—Cat. No:210212). This technique generates DNA of four different sizes specific to each serotype from viral RNA which are visualized by agarose gel electrophoresis. RNA extraction was done using NEB RNA extraction kit (Cat No: T2010S). The steps are elaborated in S1 Text (Section S1.3 and Tables A to F.

## Partial Genome Sequencing and phylogenetic analysis

During the step of RT-PCR of DENV genome a partial fragment from C-prM gene of 511 bp length was amplified. After purification with GeneJET PCR Purification Kit (Catalog number: K0701), Sanger sequencing was performed using ABI sequencing platform to determine nucleotide sequences of the 511 bp long PCR products of five samples (Id:1,2,3, 17 & 14). The chromatograms were viewed in FinchTV software and low-quality reads were trimmed from both ends of each sequence. BLAST (Basic Local Alignment Search Tool) of NCBI (National Center for Biotechnology Information) was utilized to confirm that the sequences belonged to DENV-2 and DENV-3 serotypes. Later, two phylogenetic dendrograms were created for these serotypes. Partial C-prM gene sequences of the isolates of this study were aligned to same sequence of known genotypes reported previously which has been used widely as an alternative to envelope protein gene [15–18]. These sequences were retrieved from the GenBank database of NCBI.

The dataset of DENV-2 and DENV-3 are shown in S1 Text (Section S1.4 and Tables G and H. A total of 57 and 62 sequences were used for the phylogenetic analysis for DENV2 and DENV-3 respectively. Isolates of genotype IV (DENV-3) could not be included due to lack of C-prM sequences of this genotype.

The phylogenetic dendrograms of the C-prM gene were constructed using the maximum-likelihood method with the MEGA6 software package (https://megasoftware.net). Multiple alignment was performed using MUSCLE algorithm [19]. The best substitution model for both DENV-2 and DENV-3 datasets were identified to the Kimura two-parameter model with invariable positions and Gamma distributed rates among sites [20]. Based on the findings of a previous study, the tree was statistically supported by bootstrapping with 500 replicates [21].

To look for common sites of mutation, partial c-prM gene sequences of this study and their closely related ones in the cladograms were aligned to reference genomes of the respective serotype. Jalview software was used to visualize the alignments and find the common mutation sites. Partial C-prM gene sequences of representative five samples were deposited to GenBank database under accession number **(OR726315-19)** (Table K in S1 Text).

## Statistical analysis

Data were analyzed using GraphPad prism version 9.3.1 (San Diego, CA, USA). The categorical variables are showed as frequencies and percentages after comparison with Chi-square test/ Fisher's exact test (two sided). Continuous variables were compared using Mann-Whitney U test (two tailed). The quantitative data are presented as median ± Inter Quartile Range (IQR) or mean ± (standard deviation) SD. A probability value of p < 0.05 was considered statistically significant.

## Results

### Participant enrollment profile of the study

A total of 2,114 suspected dengue patients were screened for enrollment and 2,017 met the inclusion criteria (Fig 1). Out of them 39 developed expanded dengue syndrome and thus were excluded. The final enrollment number was 1,978. This cohort were classified into group B (1580, 80%) and group C (398, 20%) according to case definitions following national guideline [14].

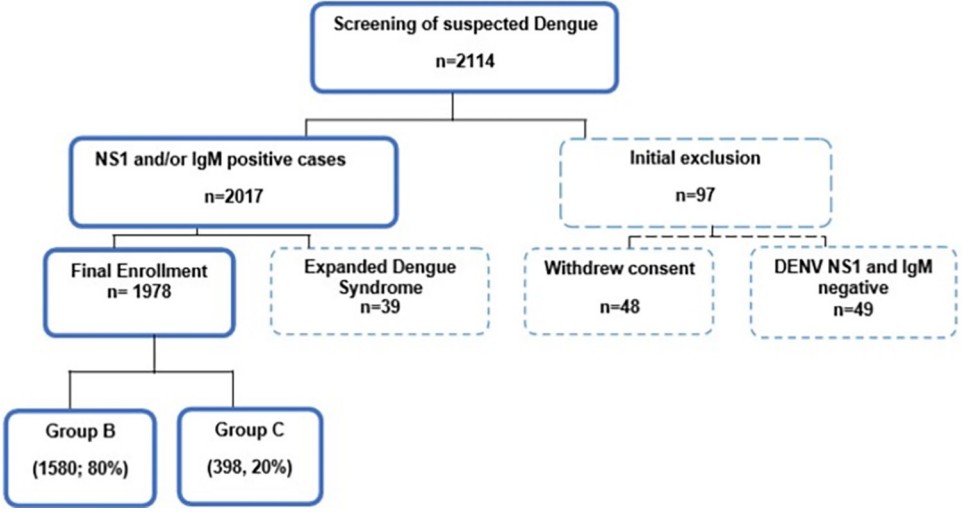

**Fig 1. Flowchart showing sequence of patient enrollment.** NS1- Non-structural protein 1; IgM- Immunoglobulin M.

### Comparison between baseline characteristics and outcome between group B and group C cases

**Demographic profile and risk factor.** The median (IQR) age of the patients were 26(20 to 35). Maximum proportion of the enrolled were male (72.3%) (p = 0.0002) (Table 1). The cases were mainly residing in Dhaka (1299; 65.7%) although a higher proportion that came from outside Dhaka were observed in group C than group B (37.2% vs 26.8%). Most of the cases were students (542; 27.4%) followed by housewives (535, 26.5%), service holders (311, 15.7%) and others. Hypertension (90; 4.5%) was the most common associated co morbidity along with diabetes mellitus (DM) (70, 3.5%) and chronic kidney disease (CKD) (10; 0.5%).

**Table 1. Demographic profile and risk factors of dengue cases.**

| Characteristics: | Total (n = 1978) | Group B (n = 1580) | Group C (n = 398) | p value* |
|---|---|---|---|---|
| Age (median, IQR) | 26, 20–35 | 27, 20–36 | 25, 19–35 | **0.0053** |
| Male, n (%) | 1430 (72.3) | 113 (74.2) | 257 (64.6) | **0.0002** |
| Residence, n (%) | | | | 0.1790 |
| Dhaka | 1299(65.7) | 1049(53) | 250(62.8) | |
| Elsewhere | 679(34.3) | 531(26.8) | 148(37.2) | |
| Occupation, n (%) | | | | 0.7132 |
| Student | 542(27.4) | 441(27.9) | 101(25.4) | |
| Housewife | 525(26.5) | 418(26.5) | 107(26.9) | |
| Service holders | 311(15.7) | 250(15.8) | 61(15.3) | |
| Businessmen | 306(15.5) | 244(15.4) | 62(15.6) | |

*(Continued)*

**Table 1.** (Continued)

| Characteristics: | Total (n = 1978) | Group B (n = 1580) | Group C (n = 398) | p value* |
|---|---|---|---|---|
| Co-morbidities, n (%) | | | | 0.1950 |
| Hypertension | 90(4.5) | 78(4.9) | 12(3) | |
| Diabetes mellitus | 70(3.5) | 56(3.5) | 14(3.5) | |
| Chronic Kidney Disease | 10(0.5) | 10(0.6) | 0(0) | |

*Mann-Whitney and Chi-square test (two-sided) was applied to identify the level of significance. P-value <0.05 was considered statistically significant (bold).

## Clinical presentation and outcome

The median duration of fever was 4 days in all groups. The classical presentation of both groups were with headache (67.2%) and bodyache (76.8%). Notably, bodyache was more frequently observed in group B (1276; 80.8%) (p<0.0001). Common mode of bleeding was melaena (235; 12%). Hemorrhagic manifestations were more prevalent in group C cases than group B (167; 42% vs 588; 37.2%). Gum bleeding was observed more in group C than group B (9.5% vs 6; p = 0.0132). Both of the groups showed a high rate of GI manifestations (99.9%) (Table 2).

**Table 2. Clinical presentations and outcome of dengue cases.**

| Characteristics | Total (n = 1978) | Group B (n = 1580) | Group C (n = 398) | p value* |
|---|---|---|---|---|
| **Duration of fever (Median, IQR)** | 4, 2–6 | 4, 2–6 | 4,1–6 | 0.2443 |
| **General symptoms, n(%)** | | | | |
| Body ache | 1519(76.8) | 1276(80.8) | 243(61.0) | **<0.0001** |
| Headache | 1330(67.2) | 1054(66.7) | 276(69.3) | 0.3392 |
| Rash | 326(16.5) | 248(15.7) | 78(19.6) | 0.0694 |
| Retro orbital pain | 156(7.9) | 212(13.4) | 78(19.6) | **0.0025** |
| **Gastrointestinal manifestations, n(%)** | 1977(99.9) | 1579 (99.9) | 398(100.0) | 1.0000 |
| Vomiting | 1068(54.0) | 831(52.6) | 237(59.5) | **0.0133** |
| Nausea | 844(43.0) | 690(44.0) | 154(38.7) | 0.0788 |
| Diarrhea | 514(26.0) | 373(23.6) | 141(35.4) | **<0.0001** |
| Abdominal pain | 284(14.4) | 212(13.4) | 72(18.0) | **0.0203** |
| **Hemorrhagic manifestations, n(%)** | 755(38.2) | 588(37.2) | 167(42.0) | 0.0836 |
| Melaena | 235(12.0) | 188(12.0) | 47(12.0) | 1.0000 |
| Gum bleeding | 132(6.7) | 94(6.0) | 38(9.5) | **0.0132** |
| Conjunctival bleed | 55(2.8) | 48(3.0) | 7(1.8) | 0.2306 |
| Per vaginal bleed | 50(2.5) | 37(2.3) | 13(3.3) | 0.2862 |
| Hemoptysis | 31(1.6) | 22(1.4) | 9(2.3) | 0.2555 |
| **Signs, n(%)** | | | | |
| Positive tourniquet test | 259(13.0) | 196(12.4) | 63(15.8) | 0.0805 |
| Pleural Effusion | 96(4.8) | 80(5.0) | 16(4.0) | 0.4355 |
| Ascites | 69(3.5) | 59(3.7) | 10(2.5) | 0.2852 |
| Temperature (Median, IQR) | 100, 98–102 | 100, 98–102 | 101, 99–103 | **0.0001** |
| Heart rate (Median, IQR) | 80, 74–88 | 80, 72–88 | 82, 76–92 | **0.0002** |
| Pulse pressure (Median, IQR) | 30, 30–40 | 40, 30–40 | 20, 20–20 | **<0.0001** |
| **Outcome**[**], **n(%)** | | | | 0.2618 |
| Death | 13(0.7) | 12(0.8) | 1(0.3) | |

*: Calculated using Mann-whitney (two tailed) and Fisher's exact test (two-sided)

**: hospital outcome, *p-value <0.05 was considered statistically significant (bold).*

Vomiting, diarrhea and abdominal pain were significantly high in group C (P = 0.0133, <0.0001 and 0.0203 respectively). A total of thirteen patients (0.7%) died which mostly belonged to group B (12, 0.8%) (p = 0.2618). There was no significant difference in the physical signs among both groups, except in vitals which can be explained by the disease course.

## Hematological and biochemical differences

In group C cases low hemoglobin and hematocrit was observed with high AST (p 0.004, 0.006 and 0.0016 respectively) (Table 3). There was no statistically significant difference between the groups in terms of total WBC count, platelet count, neutrophil lymphocyte ratio, HbA1c, alanine aminotransferase (ALT), serum creatinine, RBS, serum electrolytes, serum calcium and serum lipase.

**Table 3. Comparison of hematological and biochemical characteristics between the groups (Median, IQR).**

| Characteristics (units) | Total (n = 1978) | Group B (n = 1580) | Group C (n = 398) | p-value* |
|---|---|---|---|---|
| Hemoglobin (g/dl) | 13.3,12–14.5 | 13.4,12–14.6 | 12.9,12–14.3 | **0.0040** |
| Hematocrit (%) | 40,36.7–43 | 40.1,37–43 | 39,36–43 | **0.0060** |
| Total count WBC (/μL) | 5000,3900–7000 | 5045,3993–7006 | 5000,3800–7000 | 0.4849 |
| Platelet Count (/L) | 104500,59000–156000 | 103000,59000–156000 | 106000,60000–155000 | 0.6563 |
| N:L ratio | 1.8,1–3.2 | 1.8,1–3.1 | 2,1.1–3.6 | 0.1057 |
| AST (U/L) | 100,61.3–167.3 | 93,58–150 | 130,93–200 | **0.0016** |
| ALT(U/L) | 62.5,40–110 | 61,40–106.8 | 66,43–130.5 | 0.9240 |
| Serum creatinine (μmol/L) | 1, 0.9–1.2 | 1, 0.9–1.2 | 1, 0.8–1.1 | 0.4300 |
| RBS (mmol/L) | 6.9,5.8–8.7 | 7,5.9–8.8 | 6.5,5.4–8.1 | 0.2505 |
| HbA1c (%) | 7.2,7.2–8.4 | 7.2,7.2–8.7 | 7,6.7–7.6 | 0.4545 |
| Serum Sodium (mmol/L) | 137,134–140 | 137,134–140 | 136,133.8–139.3 | 0.5813 |
| Serum Potassium (mmol/L) | 4,3.7–4.5 | 4,3.7–4.5 | 4,3.6–4.8 | 0.4765 |
| Serum Chloride (mmol/L) | 101,98–103 | 101,98–103 | 101,98–105 | 0.3898 |
| Serum Bicarbonate (mmol/L) | 24,20–28 | 24,20.5–27.5 | 24,20–28 | 0.2634 |
| Serum Calcium (mmol/L) | 8,7.4–8.9 | 8,7.5–8.6 | 8.5,7.3–9.3 | 0.3966 |
| Serum Lipase (mmol/L) | 222.5,145–484.3 | 273,157.2–615.5 | 199,110.5–210 | 0.2253 |
| Secondary infection**, n(%) | 110 (5.6) | 89 (5.6) | 21 (5.3) | 0.9026 |

*Mann-Whitney test was applied to identify the level of significance

**Secondary infection concluded by Antidengue IgG positivity

p-value <0.05 was considered statistically significant (bold).

## Comparison of management received by two groups

Enrolled patients were admitted for a median (IQR) of 5 (4 to 6) days (Table 4). The only statistically significant difference in management in both groups were in total fluid requirement and platelet transfusion. Patients in group C needed more IV fluids (5 vs 4 L) than group B (P<0.0001). Platelet transfusion was done more in Group B patients compared to Group C (2.8% vs 1.3%) (p = 0.0070). Most patients were managed by crystalloids (1694; 85.6%) and only 14 patients (0.7%) needed colloids. However, around 5% patients required both (P = 0.7872). Transfusion of fresh whole blood and FFP was done in 5.3% (104) and 0.05% (1) cases respectively. The median (IQR) requirement of blood was of 2 (1 to 2) units (P = 0.6510) which was indifferent in both groups. Only one patient (0.05%) received antibiotic in group B. No one got steroids in any of the three study sites. Only 13 (0.7%) of the patients needed ICU support mostly belonging to group B (11, 0.7%) (p = 0.6691).

**Table 4. Management delivered to both groups during the 2019 outbreak.**

| Characteristics (Median, IQR) | Total (n = 1978) | Group B (n = 1580) | Group C (n = 398) | p-value* |
|---|---|---|---|---|
| Days of hospital stay(days) | 5,4–6 | 5,4–6 | 5,4–7 | 0.8499 |
| Total IV Fluid requirement (L) | 4.5,2–6 | 4,2–6 | 5,3–7 | **<0.0001** |
| Units of whole blood transfusion (L) | 2,1–2 | 2,1–2 | 2,1–2 | 0.0651 |
| Need of IV fluid, n (%) | 1908(96.5) | 1501(95) | 395(99.2) | 0.0518 |
| Type of IV fluid given | | | | 0.7872 |
| Crystalloid, n (%) | 1694(85.6) | 1342(85) | 352(88.4) | |
| Colloid, n (%) | 14(0.7) | 11(0.7) | 3(0.8) | |
| Both, n (%) | 97(4.9) | 74(4.7) | 23(5.8) | |
| Blood transfusion, n (%) | 104(5.3) | 82(5.2) | 22(5.5) | 0.6510 |
| Apheresis platelet transfusion, n (%) | 49(2.5) | 44(2.8) | 5(1.3) | **0.0070** |
| Fresh frozen plasma, n (%) | 1(0.1) | 1(0.1) | 0(0.0) | 0.6157 |
| Use of antibiotics, n (%) | 1(0.1) | 1(0.1) | 0(0.0) | 0.6157 |
| Use of steroid, n (%) | 0(0.0) | 0(0.0) | 0(0.0) | N/A |
| ICU support, n (%) | 13(0.7) | 11.0(0.7) | 2(0.5) | 0.6691 |

*Mann-Whitney U test (two tailed) and Chi-square test (two-sided) was applied to identify the level of significance. p-value <0.05 was considered statistically significant (bold).

## Dengue serotyping and comparison of manifestations

Among 86 extracted RNA samples 81 was successfully serotyped out of which DENV-3, DENV-2 and co infection with both DENV-2 and 3 were detected in 64 (79%), 6 (7.4%) and 11 (13.58%) samples respectively (Fig 2). The gel electrophoresis images of samples, from

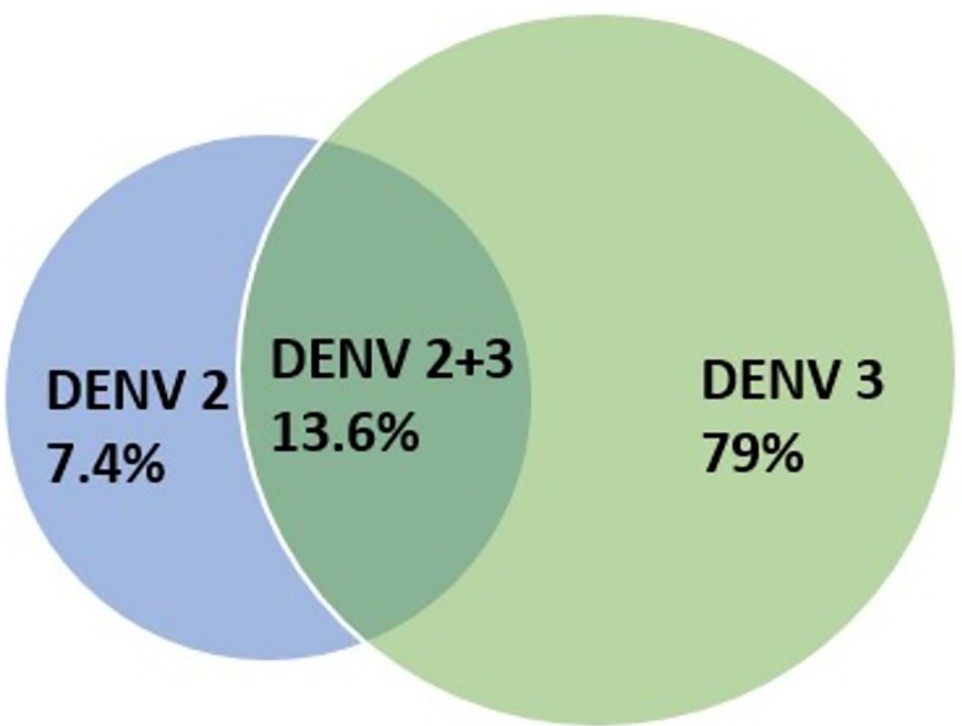

**Fig 2. Venn diagram showing percentage of prevalent serotypes during 2019 outbreak in Dhaka, Bangladesh.**

which serotypes were identified are provided in S2 Fig. Patients who were found to have DENV-3 infection or co-infection (DENV-2 &3) mostly fell into group B (>60%). On the other hand, DENV-2 was identified in equal cases of group B and Table I in S1 Text (p = ns).

Features like headache (66.7%), myalgia (50%), joint pain (50%), retro orbital pain (33.3%), and rash (33.3%) were the common complaints in these cases more frequently seen in co infected patients (DENV-2 &3) (Table J in S1 Text). Lethargy was a significant presentation in co infection group (p = 0.0314). The classical dengue symptoms like rash and retro orbital pain were only observed in about one third cases in DENV-2 and DENV-3 infections. Bleeding manifestations were more common in DENV-2 than DENV-3 (66.7% vs 51.6%). Whereas, GI manifestations were more common in DENV-3 than DENV-2 (92.2% vs 83.3%). Major mode of haemorrhage in these cohort was GI bleeding in the form of melaena (66.7%), especially in DENV-2 cases. Although negligible, genitourinary bleeds (per vaginal and hematuria) were reported in co infected cases. DENV-3 infected cases presented with bleeding involving almost all systems (including hemoptysis, gum bleeds, epistaxis, IV puncture bleeds etc). Although these differences were not statistically significant.

## Sequence analysis and phylogenetic relationship

In addition to serotyping, the nucleotide sequences of 511 bp fragment of partial C-prM gene of sample 1, 2,3,17 and 14 were identified (GenBank accession number OR726315-19 respectively). The sequences are listed in Table K in S1 Text. NCBI BLAST was utilized on samples 1, 2, 17 and 14 (sample 3 was excluded due to small sequence size) to confirm whether these sequences belonged to serotypes determined by PCR and Gel Electrophoresis.

Regarding DENV-3 serotype, the sequence from Sample no 1(OR726315) showed 99.43% identity with DENV-3 isolates that were reported previously to circulate in Dhaka city during the year 2017 [15]. Sample no 2(OR726316) was found to be identical (100% similarity) with isolates (MW599418.1, MW599415.1) prevalent during 2019 outbreak in Dhaka and Rangpur city of Bangladesh [18]. The sequence from sample no. 17(OR726318) exhibited 100% identity with two GenBank entries (MN922033.1, MN922034.1) of DENV-3 that were isolated in China from Bangladeshi travelers in the year 2019. As for DENV-2 serotype, sample no. 14 (OR726319), 99.74% percent identity with three DENV-2 isolates (LC436675.1, LC436674.1, LC436673.1) was observed and all of these were reported to be present in 2017 outbreak in Bangladesh [15].

Partial C-prM gene sequences were utilized to create two separate dendrograms for each serotype. The findings were consistent with previously reported genotypes where envelope gene (E-gene) sequence was used. The DENV-3 isolates obtained in this study belong to genotype 1 as in the dendrogram, these isolates can be found clustered around the clades previously reported to be of genotype I, which is in consensus with the sequences obtained from BLAST search. (**Fig 3**). In the case of DENV-2 dendrogram, the isolate of this study was found to be in the clade of Cosmopolitan genotype (**Fig 4**). Moreover, it was found to very closely related to DENV-2 isolates that were collected from Bangladesh during the year of 2017, 2018, 2019. Among them, the sequences of 2017 have already been reported to of Cosmopolitan genotype.

## Discussion

Predominance in male (72.3%) cases was observed in this study which was a constant observation in previous outbreaks [22–24] (Table 1). This male dominance can be explained by restrictions of health seeking behavior in woman (probability is 1.73 times higher in male) [25] due to cultural and socioeconomic factors [26,27]. Only two studies, from 2011 (54%) and 2019 (57.1%), have reported female dominance [24,28]. The median (IQR) age of this cohort

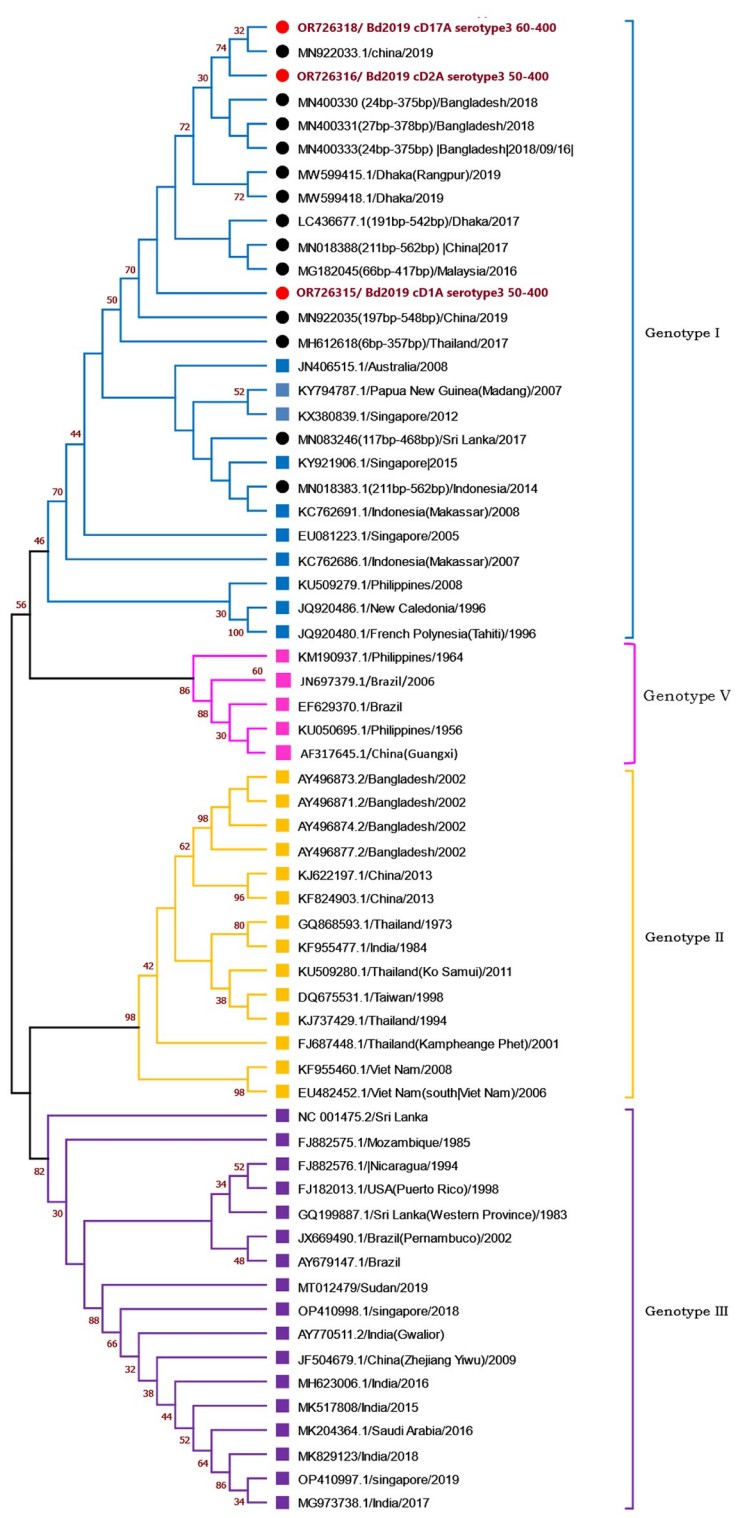

**Fig 3. Dendrogram of partial C-prM gene sequences of different DENV-3 isolates.** Red circle—sequences determined in this study, Black circle—sequences from BLAST search with no previous report about genotype. Squares: Sequences retrieved from GenBank with reports about genotypes in previous studies,[15–17]. Bootstrapping values larger than 30 are shown at the respective nodes. Respective genotypes are shown on the right side.

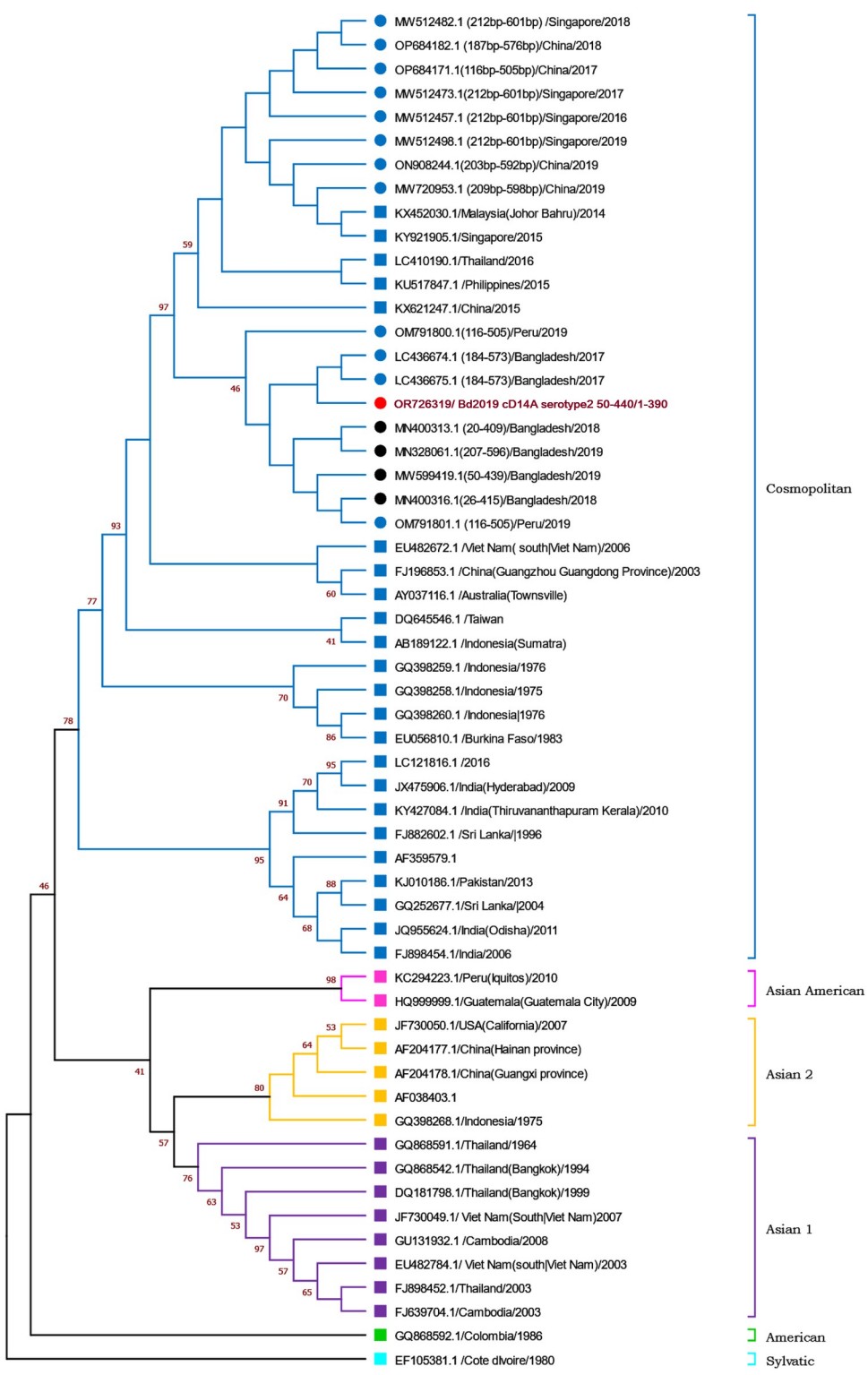

**Fig 4. Dendrogram of partial C-prM gene sequences of different DENV-2 isolates.** Red circle—sequences determined in this study, Black circle—sequences from BLAST search with no previous report about genotype. Blue circle—sequences from BLAST search that were reported as Cosmopolitan genotype. Squares: Sequences retrieved from GenBank with reports about genotypes in previous studies,[15–17]. Bootstrapping values larger than 30 are shown at the respective nodes. Respective genotypes are shown on the right side.

was 26(20 to 35). Since the outbreak of 2000, majority reports showed similar younger age distribution which may be related to their more outdoor activities [22,23,29–31]. However, in 2011 Islam et al. [28] reported a mean age of 46 years (SD 28 to 64). A few studies of 2019 have also reported prevalence in older age group (mean: 37–38 years) [24,32]. Which is in line with the global burden of disease study 2019 showing an increasing trend of incidence in elderly [33]. The occupation background and associated co-morbidities of the cases were similar to last outbreak [31,34]. Most of the cases who came from outside Dhaka fell into group C (Table 1) or dengue shock syndrome (DSS). This might indicate a delay in adequate management in this group since the cases were referred from other cities.

This study shows a decline in cutaneous manifestations from 55% to 16.5% and bleeding manifestations from 46% to 38.2% from previous studies [22,35]. Sub-conjunctival bleeds were more common (20%) in 2016 [23] but this study reported more instances of melaena (12%) (Table 2). Similar pattern was seen in 2002 (50% GI bleeds) when DENV-3 was predominant [29]. Arifuzzaman et al. reported that patients having deranged transaminases had more chance of bleeding morbidities [35]. In this study a higher AST in patients with group C (p 0.0016) or DSS been observed. Although more haemorrhagic manifestations were reported in group B. Only 5.6% of the cases were secondary infection in this cohort (Table 3). The major infecting serotypes, DENV-3 and DENV-2 was already reported to be more pathogenic and associated with severe manifestations [36,37]. This might explain the increased number of severe cases in 2019. This also bell an alarm that in near future, exposure of Dhaka residents to a different strain could bring more severe outbreaks with increased number of severe cases and death.

Concurrent infection with multiple serotypes can be transmitted by a single *Aedes aegypti* mosquito harboring multiple serotypes or multiple bites from different mosquitoes. Antibody dependent enhancement (ADE) can enhance the susceptibility to heterotypic infection as it can promote coexistence of strains that alters the dynamics of presentations [12,38]. This may be the reason behind increase prevalence of severe manifestations like fluid leakage and other warning signs in these cases [39]. However, there are other studies stating that mono infection and co infections have indifferent presentations [40]. According to previous reports primary infection with DENV-3 and secondary infection with DENV-2, DENV-3 and DENV-4 showed increased the risk of severe dengue [41]. Nisalak et al conducted a study in Thailand for near about three decades and found that DENV-3 predominance was responsible for severe outbreak years and DENV-2 was responsible for most of the secondary dengue infections and DHF [11]. In Bangladesh DENV-3 resurfaced after nine years in 2017 from the first dengue case detection in 1964 [7,42,43]. During the four decade period of its predominance, only one massive outbreak took place in 2000. Back then concurrent infection (12.5%) with DENV-2 and DENV-4 with DENV-3 was recorded [4]. After 2000 outbreak, this study of 2019 reports one of the first evidence of concurrent infection with multiple serotypes (DENV-2 and 3) (Fig 2) and their clinical profile from three tertiary care centers of Dhaka city. Although from previous report it is known that DENV-3 Genotype II has been circulating in the continent [29], this study marks the presence of DENV-3 (genotype-I) which is in line with Titir et al. [18]. Although whether there has been a complete shift of the strain that has resulted this mass outbreak needs much more exhaustive sero-surveillance.

This study shows no significant difference of infecting serotype amongst Group B and C cases (Table I in S1 Text). In this cohort, DENV-3 mostly presented with GI complaints which is in line with Halsey et al. [44]. Previous studies recorded that joint pain are common presentations of DENV-3 and DENV-2 [44,45]. This interesting shift of symptoms (musculoskeletal to gastro-intestinal) in recent outbreaks is a future area of research. Significant relationship of low platelet count with DENV-2 was also reported previously [45] but we only found higher

frequency of bleeding manifestations (Table J in S1 Text). The shifts in clinical presentations can also be observed in other countries. In India, DENV-3 was a predominant infecting serotype in 2004 and from 2016 to 2018[46–48]. Interestingly, increase in GI symptoms were prevalent around those times with spike in severe dengue cases [49,50]. Similar pattern can be seen in Sri Lanka during 2017 to 2018[51] when in late 2016 DENV-3 was found co circulating with dominant DENV-2(cosmopolitan)[52]. Although ADE is a factor that enhances heterotypic infection by promoting coexistence of multiple serotypes [12,38], most of the cases in this study was of primary infection as per RDT reports. Unfortunately, further correlation with antibody titres were not done due to poor resource setting.

The detected DENV-3 (Genotype I) isolate (OR726315) was found in closer relationship with isolates from 2017 (Dhaka) and 2019 (Dhaka and Rangpur) from Bangladesh. It was also found to be related to one isolate from China from 2019 which was collected from Bangladeshi travelers (Fig 3). DENV-2 (Cosmopolitan) isolate identified in this study showed connection with reported isolates of 2017–2019 from the same region. It may also be related to isolates of Peru from 2019 (Fig 4). This suggests there were no major molecular shift in these isolates from 2017 to 2019.

The recent studies and national case management guideline recommend against using steroids and unnecessary antibiotics in dengue as there is no evidence of usefulness [53]. In this cohort of 1978 people only one Group B case received antibiotics and none of them received steroids (Table 4). This picture ascertains prompt adherence to the national guideline in 2019 outbreak in these study sites [14]. ICU transfers and mortality rate was also minimal than other hospital reports of the same outbreak [32].

The mortality rate, as reported by DGHS (Director General Health Services), Bangladesh press release, is on uptrend from 0.16% in 2019 to 0.36% in 2021. In the year 2022, it has levelled up to 0.45% [9]. This number is just the tip of the ice berg due to the passive surveillance system and inadequate lab facilities. After, 2019 outbreak there has been a revision in national guideline and government has ensured proper training to health care providers to ensure timely treatment.

In this study, serotyping of all the samples could not be done due to low resources. More engaging studies involving other regional areas need to be done in light of this work for better understanding of viral evolution and its correlation with clinical and laboratory dynamics. This way we can probably anticipate the nature of future outbreaks and detect loopholes in the tackling system.

## Conclusion

In order to forecast the severity of an outbreak in a hyperendemic country like Bangladesh, type-specific DENV records must be documented. Because of global trade, increased cross country mobility, and rapid and unplanned urbanization emergence of novel strains and co-circulation of several strains are a significant problem in Bangladesh. Focused measures need to be taken to enhance laboratory capacity, strengthening sero-surveillance and reducing the out-of-pocket expenditure by implementing guideline based management and evaluation.

## Supporting information

**S1 Fig. Enrollment profile of three study sites in Dhaka, Bangladesh.** The red triangle represents the study co-ordinator site. The orange, blue and green circle depicts three data collection points. Site wise number of enrollment (black brackets) and percentage of group B and group C (purple and red respectively) are also shown. Map was plotted using Datawrapper

website (available at https://www.datawrapper.de/_/2ew1H/). Basemap shapefile was extracted from OpenStreetMap with due permission from the concerned. (Open Database License, ODbL 1.0, see https://www.openstreetmap.org/copyright").
(TIF)

**S2 Fig. (S2A and S2B):** Figure showing specimens from gel electrophoresis of DNA fragments generated at step 3. Fragment size of 290bp and 119bp suggests presence of DENV-3 and DENV-2 respectively. Concurrent infection by DENV-2 and DENV-3 can also be seen in S2B Fig. Presence of no other serotype was identified.
(TIF)

**S1 Text. Additional supplement of methodology and results including supporting information Tables A to K.** Table A: List of primers used in PCR reactions. Table B: Reaction condition for reverse transcriptase PCR. Table C: Reagent composition for reverse transcriptase PCR. Table D: Reaction condition of nested PCR. Table E: Reagent composition of Nested PCR. Table F: Serotypes and respective DNA fragment sizes. Table G: List of DENV-2 sequences used in the dendrogram. Table H: List of DENV-3 sequences used in the dendrograms. Table I: Serotype distribution between group B and C cases out of 81. Table J: Description of clinical presentations and laboratory parameters of 81 cases according to serotype distribution during the 2019 outbreak. Table K: Partial sequence of sample 1, 2, 3, 17 & 14.
(DOCX)

## Acknowledgments

To the doctors and healthcare workers of Dhaka Medical College and Hospital, Mugda Medical College Hospital and Sir Salimullah Medical College Hospital, for their relentless efforts in managing these case and providing support to conduct this study.

## Author Contributions

**Conceptualization:** Fazle Rabbi Chowdhury, Md Mujibur Rahman.

**Data curation:** Fazle Rabbi Chowdhury, Zazeba Hossain, Forhad Uddin Hasan Chowdhury, Mohammad Anwarul Bari, Sudip Ranjan Deb, Mustak Ibn Ayub, Md Uzzwal Mallik, Sakib Aman, Mohammad Ahsanul Haque, Md Arman Hossain, Murada Alam, Muhammad Kamrul Islam.

**Formal analysis:** Fazle Rabbi Chowdhury, Zazeba Hossain, Nahid Parvez.

**Funding acquisition:** Md Mujibur Rahman.

**Investigation:** Fazle Rabbi Chowdhury, Forhad Uddin Hasan Chowdhury, Mohammad Anwarul Bari, Sudip Ranjan Deb, Md Uzzwal Mallik, Sakib Aman, Mohammad Ahsanul Haque, Md Arman Hossain, Murada Alam, Muhammad Kamrul Islam.

**Methodology:** Fazle Rabbi Chowdhury.

**Project administration:** Fazle Rabbi Chowdhury, Forhad Uddin Hasan Chowdhury, Md Mujibur Rahman.

**Resources:** Fazle Rabbi Chowdhury, Mohammad Anwarul Bari, Sudip Ranjan Deb, Md Uzzwal Mallik, Md Mujibur Rahman.

**Software:** Zazeba Hossain, Nahid Parvez.

**Supervision:** Fazle Rabbi Chowdhury, Md Mujibur Rahman.

**Validation:** Fazle Rabbi Chowdhury, Md Mujibur Rahman.

**Visualization:** Fazle Rabbi Chowdhury, Md Mujibur Rahman.

**Writing – original draft:** Fazle Rabbi Chowdhury, Zazeba Hossain, Nahid Parvez.

**Writing – review & editing:** Fazle Rabbi Chowdhury, Zazeba Hossain, Md Mujibur Rahman.

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
