## [Decision Letter · Decision Letter 0]

11 Apr 2024

Dear Dr. Chowdhury,

Thank you very much for submitting your manuscript "Clinical and laboratory comparison of severe (Group B and C) Dengue cases with molecular characterization from 2019 epidemics in Dhaka, Bangladesh" for consideration at PLOS Neglected Tropical Diseases. As with all papers reviewed by the journal, your manuscript was reviewed by members of the editorial board and by several independent reviewers. In light of the reviews (below this email), we would like to invite the resubmission of a significantly-revised version that takes into account the reviewers' comments. 

The Authors are expected to address all the criticisms by all Reviewers. In particular, please provide a clear definition and clarify the ascertainment for primary and secondary infections (Reviewers #1, #2 and #3), clarify how fluid requirement was calculated and provide more explanation for Table 2 results (Reviewer #2), explain and assess the impact of recruitment criteria, further describe and discuss Table 3 results especially hemorrhagic and GI manifestations, and move some of the results from the supplementary to the main text, especially for patients characteristics, (Reviewer #3). In additional to these comments, please address:

1. Copy editing is needed

2. Abstract, “forecast future changes in the clinical or serological profile of this virus”. While I agree that the study may help future planning, the study analysis did not forecast changes and it does not appear to be a clear study purpose. Please strengthen the relevant analysis and discussion, or alternatively remove in the abstract and revise accordingly in the main text,.

3. Tables 1-3. Please use standard symbols for the footnotes

4. Tables 1-3. Please state the meaning of the bolded numbers (p-values and numbers/%)

5. Table 2. Please remove the level of significance in the footnote.

6. Table 1 & 3. Please provide the units directly in the table.

We cannot make any decision about publication until we have seen the revised manuscript and your response to the reviewers' comments. Your revised manuscript is also likely to be sent to reviewers for further evaluation.

Sincerely,

Eric HY Lau, Ph.D.

Academic Editor

Andrea Marzi

Section Editor

The Authors are expected to address all the criticisms by all Reviewers. In particular, please provide a clear definition and clarify the ascertainment for primary and secondary infections (Reviewers #1, #2 and #3), clarify how fluid requirement was calculated and provide more explanation for Table 2 results (Reviewer #2), explain and assess the impact of recruitment criteria, further describe and discuss Table 3 results especially hemorrhagic and GI manifestations, and move some of the results from the supplementary to the main text, especially for patients characteristics, (Reviewer #3). In additional to these comments, please address:

1. Copy editing is needed

2. Abstract, “forecast future changes in the clinical or serological profile of this virus”. While I agree that the study may help future planning, the study analysis did not forecast changes and it does not appear to be a clear study purpose. Please strengthen the relevant analysis and discussion, or alternatively remove in the abstract and revise accordingly in the main text,.

3. Tables 1-3. Please use standard symbols for the footnotes

4. Tables 1-3. Please state the meaning of the bolded numbers (p-values and numbers/%)

5. Table 2. Please remove the level of significance in the footnote.

6. Table 1 & 3. Please provide the units directly in the table.

Reviewer's Responses to Questions

**Key Review Criteria Required for Acceptance?**

**Methods**

-Are the objectives of the study clearly articulated with a clear testable hypothesis stated?

-Is the study design appropriate to address the stated objectives?

-Is the population clearly described and appropriate for the hypothesis being tested?

-Is the sample size sufficient to ensure adequate power to address the hypothesis being tested?

-Were correct statistical analysis used to support conclusions?

-Are there concerns about ethical or regulatory requirements being met?

Reviewer #1: The project titled "Clinical and laboratory comparison of severe (Group B and C) Dengue cases with molecular characterization from 2019 epidemics in Dhaka, Bangladesh" had very clear objectives to describe the clinical and

laboratory variations among severe dengue cases. The authors also conducted a serotype survey to enable them predict future changes in the clinical or serological profile of this virus. The population studied was from a dengue fever outbreak that happened in 2019. The sample size was sufficient. The clinical and laboratory evaluations were appropriate.

Reviewer #2: How were the cases classified as primary or secondary infection.Details not provided.

Reviewer #3: (No Response)

**Results**

-Does the analysis presented match the analysis plan?

-Are the results clearly and completely presented?

-Are the figures (Tables, Images) of sufficient quality for clarity?

Reviewer #1: The results were clearly presented and they did match the analysis plan. The quality of the figures are good and clear.

Reviewer #2: in Table 2 fluid requirement was calculated according to what criteria? Also is this requirement per 24 hours?

In types of fluid only 84.9 % and 88% in group B and groub C received fluid and 15 % didnt receive any crystalloid which is the first fluid given in patients with preshock or shock.Kindly explain this.Also 49 patients received platelet transfusion more in group B rather than group C which showed more hemorrhagic menifestations as quoted in table 3.

Reviewer #3: (No Response)

**Conclusions**

-Are the conclusions supported by the data presented?

-Are the limitations of analysis clearly described?

-Do the authors discuss how these data can be helpful to advance our understanding of the topic under study?

-Is public health relevance addressed?

Reviewer #1: The authors concluded that emergence of novel strains and co-circulation of several strains are a significant problem in Bangladesh which could predispose to more severity of cases in the future. The limitation of serotyping all samples due to limited resources was mentioned. The public health importance of the study was mentioned.

Reviewer #2: yes

Reviewer #3: (No Response)

**Editorial and Data Presentation Modifications?**

Reviewer #1: My suggestions are below:

Abstract

1. Sentence: “The study enrolled 1978 participants who were grouped into group B and C according to national guidelines”. 

Issue – Define what is meant by group B and C 

2. Sentence: “Median (IQR) of the cohort was 26(11 to 41)”

Issue – median of what? Besides, it seems it is a repetition.

3. Sentence: “Only 5.56% were secondary infections”

Issue: Round up 5.56% to one decimal point as precision measurement is not required here.

4. Sentence: “Only 13 (0.66%) patients succumbed to death …..”

Issue: State the number of patients without qualifying as ‘only’ and round up all decimals to one decimal point across the entire manuscript.

5. Sentence: “………. 12 of them belonged to group B.”

Issue: the sentences that follow should describe the characteristics of the group B who had more deaths.

6. Sentence: “Out of 81 samples 79% patients were serotyped as DENV-3, followed by

DENV 2,3 (13.6% ) and DENV-2 (7.4%).”

Issue: Do you mean, ‘Out of 81 samples collected from patients, 79% were …..”? Also, it seems DENV 2 is being reported twice. Please clarify

Methods

Study site and enrollment: It was mentioned that the study was done from 1st July to 31st December, 2019 whereas, in the author summary and abstract sections, study period was reported as August to December 2019? Please harmonize.

Results

Dengue serotyping: “the former two serotypes”. It is advisable to mention the “former serotypes” since not all readers are familiar with the former serotypes. 

• Recast this sentence for clarity: “Among the three patterns of identified serotypes majority

(43, 67.2%) of the cases belonged to group B in DENV-3 and DENV-2 and 3 co infection (7, 63.6%).”

Discussion

• It will be helpful to define ‘primary infection’ in the manuscript.

• “ICU transfers and mortality rate was also minimal than other sites [44].” Expatiate on what is meant by ‘other sites’

Reviewer #2: (No Response)

Reviewer #3: (No Response)

**Summary and General Comments**

Reviewer #1: The study described the changing clinical presentation of dengue cases which is very relevant for the global community because of the changing dynamics of vector borne diseases particularly in this era of climate change.

Reviewer #2: (No Response)

Reviewer #3: (No Response)

PLOS authors have the option to publish the peer review history of their article (what does this mean?). If published, this will include your full peer review and any attached files.

Reviewer #1: No

Reviewer #2: Yes: Somia Iqtadar

Reviewer #3: No
---

## [Decision Letter · Decision Letter 1]

6 Sep 2024

Dear Dr. Chowdhury,

Thank you very much for submitting your manuscript "Clinical and laboratory comparison of severe (Group B and C) Dengue cases with molecular characterization from 2019 epidemics in Dhaka, Bangladesh" for consideration at PLOS Neglected Tropical Diseases. As with all papers reviewed by the journal, your manuscript was reviewed by members of the editorial board and by several independent reviewers. In light of the reviews (below this email), we would like to invite the resubmission of a significantly-revised version that takes into account the reviewers' comments. 

The Authors have addressed the criticisms by the Reviewers. However, there are some remaining issues to be addressed:

1. Methods, Line 147-150 and Discussion Line 416-420. There seems to be some intention to analyze primary infections, and in the discussion “most of the cases in this study was of primary infection as per RDT test reports”. However the relevant results were missing. 

2. Please also clarify how primary and secondary infections were analyzed.

3. Table 1. Please use consistent formatting, e.g. “(113) 74.2” should be “113 (74.2)”

4. Line 233, please change “(p=<0.0001)” to “(p<0.0001)”

5. Tables 2, 3 and 4. Most of the outcomes did not have a symmetric distribution and are better presented with median and IQR. Please also update the relevant text accordingly.

6. Table 2, please use consistent decimal places for the percentages

7. Table 2, please change “>0.9999” to “1.000”

8. Table 2, please use the same decimal place (1 dp) for mean and SD.

9. Figure 2, please use 1 decimal place for percentages

10. Line 418, please present the full term of RDT at first appearance, and if the test was used it should be clearly state in the Methods.

11. Discussion, line 353-359. Please confirm if the mean age was younger than the cited studies only, or exceptionally young? In the latter case, are there any reasons explaining the younger cases in this study? 

12. Discussion, if the study population is younger than usual, the impact of a different age distribution should be accessed more carefully when comparing with previous studies (Line 365-378). The authors may consider supplementary analyses stratified by age.

We cannot make any decision about publication until we have seen the revised manuscript and your response to the reviewers' comments. Your revised manuscript is also likely to be sent to reviewers for further evaluation.

Sincerely,

Eric HY Lau, Ph.D.

Academic Editor

Andrea Marzi

Section Editor

The Authors have addressed the criticisms by the Reviewers. However, there are some remaining issues to be addressed:

1. Methods, Line 147-150 and Discussion Line 416-420. There seems to be some intention to analyze primary infections, and in the discussion “most of the cases in this study was of primary infection as per RDT test reports”. However the relevant results were missing. 

2. Please also clarify how primary and secondary infections were analyzed.

3. Table 1. Please use consistent formatting, e.g. “(113) 74.2” should be “113 (74.2)”

4. Line 233, please change “(p=<0.0001)” to “(p<0.0001)”

5. Tables 2, 3 and 4. Most of the outcomes did not have a symmetric distribution and are better presented with median and IQR. Please also update the relevant text accordingly.

6. Table 2, please use consistent decimal places for the percentages

7. Table 2, please change “>0.9999” to “1.000”

8. Table 2, please use the same decimal place (1 dp) for mean and SD.

9. Figure 2, please use 1 decimal place for percentages

10. Line 418, please present the full term of RDT at first appearance, and if the test was used it should be clearly state in the Methods.

11. Discussion, line 353-359. Please confirm if the mean age was younger than the cited studies only, or exceptionally young? In the latter case, are there any reasons explaining the younger cases in this study? 

12. Discussion, if the study population is younger than usual, the impact of a different age distribution should be accessed more carefully when comparing with previous studies (Line 365-378). The authors may consider supplementary analyses stratified by age.

Reviewer's Responses to Questions

**Key Review Criteria Required for Acceptance?**

**Methods**

-Are the objectives of the study clearly articulated with a clear testable hypothesis stated?

-Is the study design appropriate to address the stated objectives?

-Is the population clearly described and appropriate for the hypothesis being tested?

-Is the sample size sufficient to ensure adequate power to address the hypothesis being tested?

-Were correct statistical analysis used to support conclusions?

-Are there concerns about ethical or regulatory requirements being met?

Reviewer #2: objectives of the study are clearly elaborated ,with appropriate study design .sample size is sufficient and correct statistical nalysis has been applied .There are no concerns about ethical or regulatory requirements.

Reviewer #3: (No Response)

**Results**

-Does the analysis presented match the analysis plan?

-Are the results clearly and completely presented?

-Are the figures (Tables, Images) of sufficient quality for clarity?

Reviewer #2: analyses matches the plan with clearly menioned results in the form of tables and elaborated in the manuscript as well.Figures and tables quality is good.

Reviewer #3: (No Response)

**Conclusions**

-Are the conclusions supported by the data presented?

-Are the limitations of analysis clearly described?

-Do the authors discuss how these data can be helpful to advance our understanding of the topic under study?

-Is public health relevance addressed?

Reviewer #2: data supports tyhe conclusion and results are discussed in detail with refrences and comparison from previous studies of this region making them relevant to address public health .

Reviewer #3: (No Response)

**Editorial and Data Presentation Modifications?**

Reviewer #2: No further changes required.

Reviewer #3: (No Response)

**Summary and General Comments**

Reviewer #2: Its a really useful study with strong data.My only concern is use of RDT for diagnosis.In a setup where they are doing molecular characterization and serotyping absence of ELISA testing remains questionable.Authors have quoted this limitation themselves in the manuscript though.

This study provides them to delve deeper into research related to serospecific presentation of dengue cases and presence of severe disease and DHF in primary infection and ADE hypothesis for plasma leakage .

Reviewer #3: (No Response)

PLOS authors have the option to publish the peer review history of their article (what does this mean?). If published, this will include your full peer review and any attached files.

Reviewer #2: Yes: Somia Iqtadar

Reviewer #3: Yes: Dr. Faiz Ahmed Raza
---

## [Editor Report · Decision Letter 2]

4 Oct 2024

Dear Dr. Chowdhury,

Thank you very much for submitting your manuscript "Clinical and laboratory comparison of severe (Group B and C) Dengue cases with molecular characterization from 2019 epidemics in Dhaka, Bangladesh" for consideration at PLOS Neglected Tropical Diseases. As with all papers reviewed by the journal, your manuscript was reviewed by members of the editorial board and by several independent reviewers. The reviewers appreciated the attention to an important topic. Based on the reviews, we are likely to accept this manuscript for publication, providing that you modify the manuscript according to the review recommendations. 

The Authors have addressed most of the comments. However, there are some remaining minor issues around presentation of statistics to be addressed:

1. Line 106, please correct the number “1,01,358”

2. Tables 2, 3 and 4 have now presented median and a number related to IQR, which is an improvement. As point out by a reviewer in the previous round of review, IQR should be a range. This was well presented for age (IQR = 11-41) in Table 1, but not for other variables.

3. Tables 2, 3 and 4, please change “(median± IQR)” to “(median, IQR)”. As an example from Table 1, the statistics for age can be presented as “26, 11- 41”

4. Table 2, please use 1 decimal place consistently for the percentages for all variables

Sincerely,

Eric HY Lau, Ph.D.

Academic Editor

Andrea Marzi

Section Editor

The Authors have addressed most of the comments. However, there are some remaining minor issues around presentation of statistics to be addressed:

1. Line 106, please correct the number “1,01,358”

2. Tables 2, 3 and 4 have now presented median and a number related to IQR, which is an improvement. As point out by a reviewer in the previous round of review, IQR should be a range. This was well presented for age (IQR = 11-41) in Table 1, but not for other variables.

3. Tables 2, 3 and 4, please change “(median± IQR)” to “(median, IQR)”. As an example from Table 1, the statistics for age can be presented as “26, 11- 41”

4. Table 2, please use 1 decimal place consistently for the percentages for all variables

Figure Files:

Data Requirements:

Reproducibility:

References

---

## [Editor Report · Decision Letter 3]

17 Oct 2024

Dear Dr. Chowdhury,

Thank you very much for submitting your manuscript "Clinical and laboratory comparison of severe (Group B and C) Dengue cases with molecular characterization from 2019 epidemics in Dhaka, Bangladesh" for consideration at PLOS Neglected Tropical Diseases. As with all papers reviewed by the journal, your manuscript was reviewed by members of the editorial board and by several independent reviewers. The reviewers appreciated the attention to an important topic. Based on the reviews, we are likely to accept this manuscript for publication, providing that you modify the manuscript according to the review recommendations. 

The Authors have addressed some of the comments. However, there are still a few remaining minor issues that to be addressed before this manuscript can be considered for acceptance:

1. Line 106, please correct the number “1,01,354”. Was that 1 million or 100 thousand?

2. Table 2, please correct the numbers “20, 0” for the pulse pressure of Group C

3. Table 3, please add a comma in “50001900- 8100”

4. Table 3, please correct the negative values (e.g. N:L ratio, AST and others)

4. Tables 3 and 4, please harmonize the decimal places for the numbers presented

Sincerely,

Eric HY Lau, Ph.D.

Academic Editor

Andrea Marzi

Section Editor

The Authors have addressed some of the comments. However, there are still a few remaining minor issues to be addressed:

1. Line 106, please correct the number “1,01,354”. Was that 1 million or 100 thousand?

2. Table 2, please correct the numbers “20, 0” for the pulse pressure of Group C

3. Table 3, please add a comma in “50001900- 8100”

4. Table 3, please correct the negative values (e.g. N:L ratio, AST and others)

4. Tables 3 and 4, please harmonize the decimal places for the numbers presented

Figure Files:

Data Requirements:

Reproducibility:

References

---

## [Editor Report · Decision Letter 4]

8 Nov 2024

Dear Dr. Chowdhury,

We are pleased to inform you that your manuscript 'Clinical and laboratory comparison of severe (Group B and C) Dengue cases with molecular characterization from 2019 epidemics in Dhaka, Bangladesh' has been provisionally accepted for publication in PLOS Neglected Tropical Diseases.

Best regards,

Eric HY Lau, Ph.D.

Academic Editor

Andrea Marzi

Section Editor

Shaden Kamhawi

co-Editor-in-Chief

Paul Brindley

co-Editor-in-Chief

---

## [Editor Report · Acceptance letter]

18 Nov 2024

Dear Dr. Chowdhury,

We are delighted to inform you that your manuscript, "Clinical and laboratory comparison of severe (Group B and C) Dengue cases with molecular characterization from 2019 epidemics in Dhaka, Bangladesh," has been formally accepted for publication in PLOS Neglected Tropical Diseases.

Best regards,

Shaden Kamhawi

co-Editor-in-Chief

Paul Brindley

co-Editor-in-Chief
